# Trials of the Automated Particle Counter for laboratory rearing of mosquito larvae

**Mark Q. Benedict**[ID]*, **Priscila Bascuñán**[ID]◉, **Catherine M. Hunt**[ID]‡, **Erica I. Aviles**‡, **Rachel D. Rotenberry**‡, **Ellen M. Dotson**◉

Entomology Branch, Division of Parasitic Diseases and Malaria, Centers for Disease Control and Prevention (CDC), Atlanta, GA, United States of America

◉ These authors contributed equally to this work.
‡ These authors also contributed equally to this work.
* mbenedict@cdc.gov

**Data Availability Statement:** The data is available at https://osf.io/vngmf/.

**Funding:** MQB, PB, CMH, RDR and EIA are paid employees of Target Malaria, a project that receives

## Abstract

As a means of obtaining reproducible and accurate numbers of larvae for laboratory rearing, we tested a large-particle flow-cytometer type device called the 'Automated Particle Counter' (APC). The APC is a gravity-fed, self-contained unit that detects changes in light intensity caused by larvae passing the detector in a water stream and controls dispensing by stopping the flow when the desired number has been reached. We determined the accuracy (number dispensed compared to the target value) and precision (distribution of number dispensed) of dispensing at a variety of counting sensitivity thresholds and larva throughput rates (larvae per second) using < 1-day old *Anopheles gambiae* and *Aedes aegypti* larvae. All measures were made using an APC algorithm called the 'Smoothed Z-Score' which allows the user to define how many standard deviations (Z scores) from the baseline light intensity a particle's absorbance must exceed to register a count. We dispensed a target number of 100 *An. gambiae* larvae using Z scores from 2.5–8 and observed no difference among them in the numbers dispensed for scores from 2.5–6, however, scores of 7 and 8 under-counted (over-dispensed) larvae. Using a Z score ≤ 6, we determined the effect of throughput rate on the accuracy of the device to dispense *An. gambiae* larvae. For rates ≤ 98 larvae per second, the accuracy of dispensing a target of 100 larvae was - 2.29% ± 0.72 (95% CI of the mean) with a mode of 99 (49 of 348 samples). When using a Z score of 3.5 and rates ≤ 100 larvae per second, the accuracy of dispensing a target of 100 *Ae. aegypti* was - 2.43% ± 1.26 (95% CI of the mean) with a mode of 100 (6 of 42 samples). No effect on survival was observed on the number of *An. gambiae* first stage larvae that reached adulthood as a function of dispensing.

## Introduction

In the context of laboratory rearing of mosquitoes, controlling the number of larvae and the amount of food per larva determines their development rate and the size of the resulting adults [1]. In extreme cases, food per larva determines survival as well. More subtle effects on fecundity [2], susceptibility to parasites [3, 4], longevity [3] and insecticide resistance [5, 6] can also be observed as a function of size or diet level (which affects size).

core funding from the Bill & Melinda Gates Foundation and from the Open Philanthropy Project Fund, an advised fund of Silicon Valley Community Foundation to the Target Malaria project. The funds for this study were awarded to the Foundation for the Centers for Disease Control and Prevention. The funders had no role in study design, data collection and analysis or decision to publish.

**Competing interests:** The authors have declared that no competing interest exist.

There are several readily accessible methods for achieving some degree of control on the number of larvae per tray. Some are appropriate for small-scale laboratory rearing whereas others are more suitable for mass-rearing applications. For small-scale rearing, a commonly used technique is simply manually counting them into trays, but this is time-consuming and depends on the skill and diligence of the person. Other options are to estimate the larval density in comparison to photographs of known numbers in similar containers or compressing them into a particular volume [7].

When the purpose is mass-rearing, a variety of methods have been used: volumetric measures of eggs [8]; weighing [9]; and counting samples from a large volume of a well-mixed sample and calculating the volume needed to dispense the desired number [10]. Each of these has strengths and weaknesses in terms of ease and accuracy but for many purposes are suitable. A widely known device called the Complex Object Parametric Analyzer and Sorter (COPAS, www.unionbio.com/copas/) can count and sort individual larvae accurately and rapidly, and studies report no effect on survival [11]. However, it costs over US$300,000 which places it out of the reach of many laboratories unless additional analytical needs exist. The basic principle of the COPAS, optical detection of objects, is an adaptation of flow cytometry to large particles.

In this study, we tested a comparatively simplified device, the Automated Particle Counter (APC), developed by Applied Scientific Solutions, LLC (Kennesaw, GA USA), for its suitability to count and dispense mosquito larvae. The device uses changes in light intensity to detect larvae or other particles according to user parameters and controls the number dispensed by interrupting the flow of water using a pinch valve. The principle of this device is similar to that of a device reported by Mamai *et al.* [12] with some differences that are highlighted in the discussion.

The basic components of the APC consist of: a Raspberry Pi computer connected to a touch-screen display; a microcontroller; a pinch valve to regulate the stream; a motorized stirrer to distribute the larvae in the device reservoir and; a sensor and an LED light source housed in a single component Illuminator/detector. Unlike the COPAS which requires an air compressor to achieve sufficient pressure to cause water to flow through the detection chamber, the APC utilizes gravity acting on the water column (head pressure) to create flow. This limits the feasible internal diameter of the water stream since excessively small diameters reduce flow to an unusable degree, but it has the advantage that it reduces mechanical complexity.

Here, we determined the accuracy, as indicated by the difference between the number dispensed and the user-specified number, and the precision of the number dispensed by the APC, as reflected by the distribution of the data around the mean. We altered and measured various parameters of its operation including sensitivity of larvae detection, rate of larva passing the detector and the effect of the species of mosquito to determine their effect on these outcomes. Two species that are widely reared and which have different larval morphology and behavior were tested, *Anopheles gambiae* and *Aedes aegypti*. The effect of dispensing by the APC on *An. gambiae* survival to adulthood was compared with the manual counting method.

## Materials and methods

### Mosquitoes

Stocks of *An. gambiae* mosquitoes of the 'G3' strain (MRA-112) and *Ae. aegypti* of the 'New Orleans' strain (NR-49160) were obtained from the Malaria Research and Reference Resource Center (MR4, BEI Resources, Manassas VA, USA). In order to prevent unhatched embryos and shells from contaminating the larvae that were to be dispensed, *An. gambiae* embryos were hatched on 9 cm diameter filter paper discs (Whatman, GE Healthcare) supported above

the water on a platform made of cloth sponge discs (S1 Fig). This method allows nearly all larvae to wriggle off the paper into the water after hatching and a thorough removal of the eggshells can be achieved by gently lifting the paper. *Aedes aegypti* eggs were collected on seed germination paper and dried for embryonation for 4 days after collection before storage in plastic bags in a covered plastic box. They were hatched as needed by placing the germination papers with eggs in water under vacuum for 30 min. Eggs of both species were hatched in either a complex diet [13] or baker's yeast. One day after hatching, when most larvae were still in the L1 stage, they were filtered with a fine screen and gently washed with clean water. This removed all of the yeast, but removing the complex diet also required two or three rounds of decanting. Trials in which larvae were hatched with Doctors Foster and Smith Koi Staple Diet (Rhinelander, WI USA) were not successful as the food is quite flocculent and was impossible to remove by decanting.

## Description and trials of the Automated Particle Counter

The main parts of the APC are shown in Fig 1. The device is supplied with a variety of tubes, reservoirs and various spare parts including extra sensors. All of the parameters including stirring rate, LED light intensity and counting parameters can be adjusted by the user. If desired, detailed data collected by the device can be observed via a WiFi or USB connection to an external computer or mobile device, and graphic displays of the data are accessible through a web browser. Neither requires internet access. The device can be operated with all common voltages and plug types and a clear manual is provided. Before use and during its operation, the

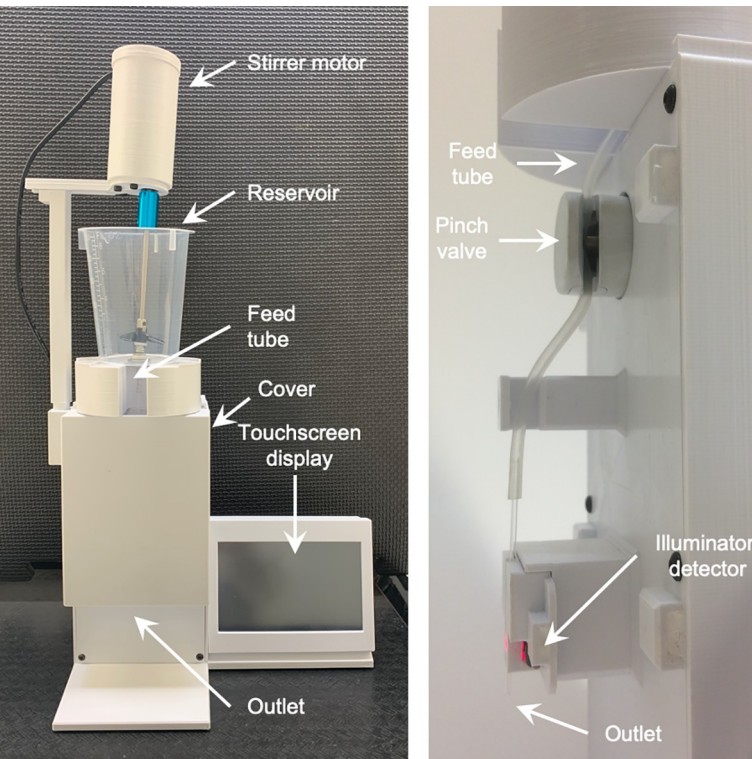

**Fig 1. The Automated Particle Counter.** The entire device viewed from the front (left) and (right) the major water stream components are shown viewed from the side with the main cover removed. The capillary shown has an ID of 1 mm. The reservoir shown contains 250 ml and the light source LED chosen was red. Other possible light options include green and infrared.

device was calibrated until the standard deviation of the device's light sensor value was < 18, which was a reasonable minimum that was consistently achieved. Note that values determined by the APC are not optical density in the standard sense.

The internal diameter (ID) of the glass capillary provided with the APC (1 mm) is sufficiently small to prevent any further drainage when the pinch valve, which is located upstream, is closed. Approximately 1.5 cm of the capillary extends below the sensor so that when the pinch valve closes, some larvae will have been counted but not dispensed and would be dispensed but not counted in the next dispensing run. In order to estimate this number, 10 direct visual observations of the number of larvae present in this section of the capillary after the pinch valve was closed were made after dispensing runs.

To understand the required speed response of the sensor in combination with the number of larvae per second (LPS) that would be passing the sensor to be dispensed, the water flow rate was determined by adding 100 ml of water to the 250 ml supply reservoir filled to either "high" or "low" levels (31 cm or 24 cm of water column above the outlet respectively) and letting it pass through the standard 1 mm ID capillary. The rates of water flow (ml/sec) at high and low levels were compared by two-sided t-test. The number of larvae being dispensed was determined from the APC data output.

**Quantifying larvae.** Less than one-day old larvae were dispensed by the APC into small plastic cups (S2 Fig) and then vacuum-filtered onto 15 cm diameter filter papers labeled with the run identification number. These were photographed using an iPhone camera (minimum 5S model) and the number of larvae was counted by at least two people. In the event of disagreement, the final count was determined by a third person who counted the larvae again and/or examined the previously marked images for discrepancies, all using the "multi-point" tool in ImageJ [14]. All trials were conducted using a target of 100 larvae and were dispensed from a 250 ml reservoir, the level of which was kept roughly at the high level (defined above) so that the main determinant of larva supply rate was the density of the larvae rather than the water flow rate.

**Z score effect on accuracy.** The APC senses changes in light intensity in real time, analyzes them and accumulates the counts using one of several algorithms chosen and parameterized by the user. The device is equipped with three algorithms: 1) 'Threshold Reset' which requires a user specified number of measurements above and below a baseline threshold to identify separate particles; 2) 'Refractory Period' is similar to 'Threshold Reset' but requires passage of a user specified amount of time between particles rather than a number of measurements. Both of these seem useful for well-separated particles but we believe would be poor at identifying overlapping particles; 3) We tested the 'Smoothed Z-Score'. This algorithm is robust since it adapts to changing water light absorbance by applying the specified number of standard deviations (Z scores) in sensed light intensity to a moving-window average baseline intensity that a particle's absorbance must exceed to trigger a 'count'. By accessing the configuration menu on the touch-screen menu, the user can select the number of standard deviations (Z score) needed to begin recording the intensity, the number of detections above the minimum to trigger a count and the number of declining values or the amount before initiating another count, which is useful to distinguish clustered individuals. These settings are stored between uses of the device so that settings do not need to be entered each time. In this study, we tested Z scores ranging from 2.5–8. The numbers dispensed at each Z score were analyzed by ANOVA and group comparisons were performed by Tukey's Honestly Significant Difference (HSD) method with Bonferroni correction. With the exception of the survival ANOVA which was analyzed using GraphPad Prism v. 8, all statistical analyses were performed using the 'Real Statistics' Microsoft Excel plugin [15] or the Excel "Analysis" plugin and an alpha of

0.05 except where corrected. Error values in parentheses following values are 95% confidence intervals of the means.

**Correlation between supply rate and accuracy.** By adding concentrated larvae or diluting them in the feed reservoir by adding water, we tested a range of LPS rates at which *An. gambiae* or *Ae. aegypti* larvae passed the sensor. The duration of each dispensing run was recorded by the APC software (S3 Fig). The relationship between LPS rates and accuracy was determined by calculating the Pearson correlation coefficient.

**Effect of APC dispensing on survival.** The effect of dispensing with the APC on *An. gambiae* survival from < 1 day old larva to the adult stage was determined by dispensing a target number of 250 larvae into each of three trays, filtering them onto discs, photographing and counting as described above before flushing them back into trays for rearing. The number of adults that resulted was compared with those obtained from three trays of 250 manually counted larvae that were prepared as follows: Larvae were manually counted by pipetting into trays in small droplets of water. They were then flushed onto filter paper using the vacuum apparatus as described above and photographs were used to make the final count of the number of larvae. In this way, both groups were subjected to vacuum filtration, photography and flushing. Two experiments (i.e. two trials of six trays each) were performed. After counting, larvae were fed on Doctors Foster and Smith Koi Staple Diet at a 0.3 mg/larvae/day rate. Prior to feeding, Koi pellets were finely ground using a Burr Mill Coffee Grinder (Black and Decker, Model CBM310BD) and passed through a 600-micron opening standard mesh sieve to remove large particles. Larvae and adult mosquitoes were reared in a room with controlled conditions of temperature and relative humidity set at 27˚C and 70% respectively, and with a regulated light: dark cycle of 12: 12 hours with 30 min of dawn and dusk to simulate natural conditions. Pupae were collected daily and their sex determined until no larvae remained. These were allowed to eclose in cages from cups of clean water after which the number of dead pupae was counted. The numbers of larvae dispensed by the APC or counted manually were compared by a two-tailed t-test. The effect of trial and counting method on the proportion of larvae that survived to the adult stage was analyzed by ANOVA considering the interaction between trial and method.

## Results

### Basic physical parameters of the APC

Because the APC is a gravity pressurized system, the water flow rate depends mainly on the level of water in the feed reservoir and any restrictions to flow in the system. Using the standard 1 mm ID capillary, we measured water flow rates when the reservoir water level was high (0.77 ml/sec; n = 6, stdev 0.012), compared to when the level was low (0.48 ml/sec; n = 7, stdev 0.007). Not surprisingly, these flow rates were different (t-test p = $7.59 \times 10^{-16}$). Since almost all of the counts were performed with the reservoir nearly at the high level, we used the higher estimate to approximate the transit time of a larva past the sensor which will be discussed further below.

Since there could be error introduced by larvae remaining in the capillary that had not been dispensed, we directly counted the number of larvae present in the capillary below the detector between successive runs at moderate LPS rates of approximately 25 LPS. In 10 runs, an average of 1.4 larvae (stdev = 1.51) were observed that had not been dispensed. In general, this should have no effect on the average accuracy of the number of larvae dispensed. However, larvae that are counted but not dispensed or dispensed but not counted contribute to the precision of the number dispensed.

Based on the LPS, size of larvae, capillary ID and water volume flow rate, it is possible to estimate the duration of the time that a larva is in the sensible view of the detector and the approximate density of larvae in the reservoir. Although a first stage larva is approximately 1 mm long (excluding setae) and could possibly be positioned laterally across a 1 mm ID capillary tube when passing the sensor, the friction of the water against the capillary wall will tend to slow the flow at the edges, an effect called hydrodynamic focusing. Based on this effect, we assume that a slow flow at the edges and the setaie help the larvae to align parallel to the sides in the faster flow at the center. We calculated that, in a 1 mm ID capillary with a water flow rate of 0.77 ml/sec, a larva moves in the tube at a speed of 980 mm/sec. This means that a ca. 2 mm long sensor window such as is equipped in the APC has ca. 2 msec during which a larva is 'visible'. We have observed that the APC sensor can make independent measurements at intervals of ca. 10 / msec (S4 Fig) and thus sensor response time is not a limiting factor in detecting larvae at the speed they passed the sensor during our trials.

One can also calculate backward to determine the concentration of larvae in the feed reservoir using the water volume flow rate per second and the LPS. Assuming perfect accuracy, at a value of 10 and 30 LPS (described further below), the concentration of larvae in the reservoir would be 13 and 39 larvae per ml respectively.

In these trials, discrepancies between the total number of larvae counted by the first two people who were enumerating larvae from photographs occurred 57.6% of the time (275 out of 477). These required reconciliation by a third person whose count was considered final. The average difference when disagreement existed was 1.76% (± 0.23).

**Z score effect on accuracy.** When using the 'Smoothed Z-Score' method, a Z score sufficiently high to ignore e.g. larval food particles and electronic noise must be selected to ensure that only larvae are counted. However, if the score is too high, numerous larvae will be dispensed but not counted, resulting in a loss of accuracy and precision as reflected in the confidence intervals (Fig 2, Table 1). At Z scores from 2.5–6, no difference in the number of larvae dispensed was observed; Z scores of 7 and 8 under-counted the number of larvae dispensed (Table 2). As a result of seeing no difference in Z scores from 2.5–6 and the manufacturer's recommendation, we routinely used a Z score of 3.5. The manufacturer's recommendation is based on a random error probability of 1/2128 data points (p = 0.00047).

**Correlation between supply rate and accuracy.** By adding concentrated larvae or diluting them with water in the feed reservoir, we tested a range of LPS rates at which *An. gambiae* or *Ae. aegypti* larvae passed the sensor. The duration of each dispensing run was recorded by the APC software (S3 Fig). The relationship between LPS rates and accuracy was determined by calculating the Pearson correlation coefficient. There was no correlation between the LPS and accuracy of numbers dispensed (Fig 3) for *An. gambiae* (p = 0.063) nor *Ae. aegypti* (p = 0.221).

**Effect of APC dispensing on survival.** Although manual counting is often considered a reference 'gold standard,' in our hands, the accuracy of manual counting when preparing this sample of six trays of 250 larvae each for the survival trials was +5% (average of 262). This, combined with the under-dispensing by the APC of -7% (n = 6, average of 233) in these trials resulted in a significant difference (df = 10, p = 0.002) in the numbers dispensed by the two methods. Survival to adulthood was high in both groups; 94.1% (± 3.58) for the manual method and 95.2% (± 2.25) for APC dispensing. There was no difference in survival from L1 to the adult stage related to the dispensing method (df = 1, p = 0.43), replicate (df = 1, p = 0.17) or their interaction (df = 1, p = 0.27).

**Overall analysis of accuracy and precision.** Considering only Z scores settings of 2.5–6 as described above, we determined the accuracy and precision of dispensing. Eliminating the values above 6 did not truncate the smaller *Ae. aegypti* data set (n = 42), but it reduced the

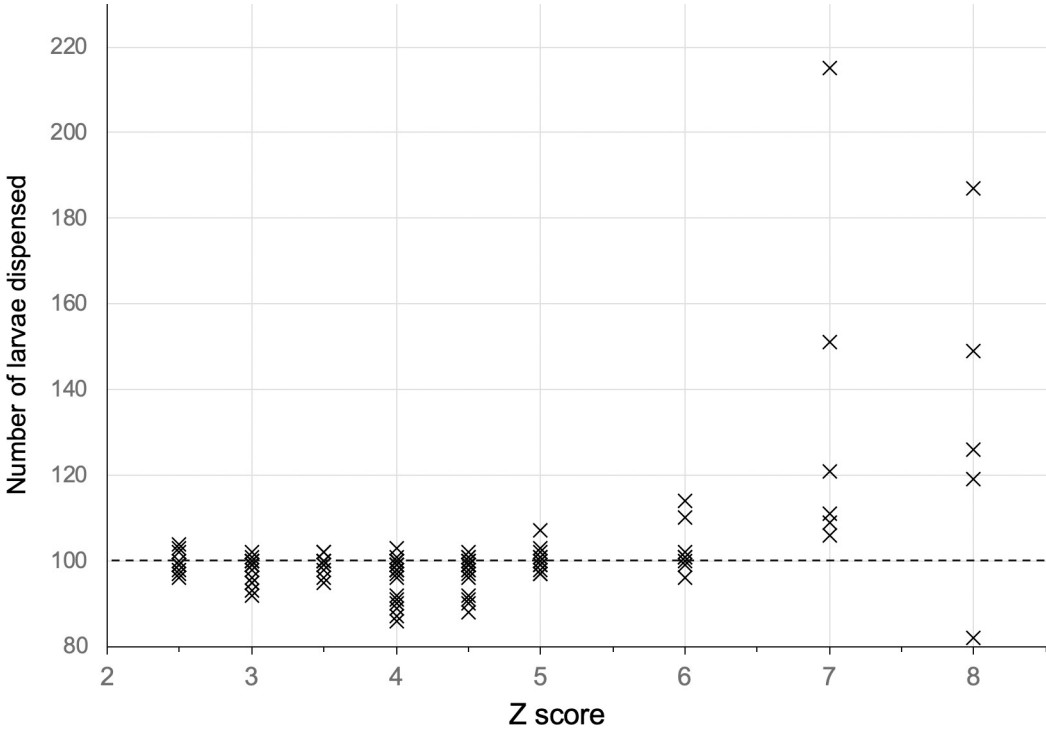

**Fig 2. Z score effect.** When using the 'Smoothed Z-Score' algorithm, a Z score sufficiently high to not detect e.g. food particles, eggshells and electronic noise must be selected to ensure that only larvae are counted. However, if the score is > 6, several larvae are not counted resulting in an excessive number being dispensed. The target number was 100 for these runs, indicated by the horizontal dashed line.

number of *An. gambiae* samples from 359 to 348. The overall accuracy of dispensing *An. gambiae* was -2.29% (± 0.72) with a mode of 99 (49 of 348 samples, Fig 4A). The accuracy of dispensing *Ae. aegypti* was - 2.43% (± 1.26) with a mode of 100 (6 of 42 samples, Fig 4B).

One could set the number to be dispensed approximately 2–3% above the desired number and compensate for accuracy. However, the distribution of the numbers dispensed is of concern and should be known to use such a dispenser effectively. The *An. gambiae* and *Ae. aegypti* datasets were analyzed separately. The *An. gambiae* counts showed significant leftward skew (Fig 4A) in the data which was not normally distributed (Shapiro Wilk p = 4.4 x 10−16).

**Table 1. Summary data of the effect of Z score on the number dispensed when 100 was specified.**

| Z score | Runs (n) | Mean (95% CI mean) | | |
|---|---|---|---|---|
| 2.5 | 10 | 99.7 | (± 1.60)[1] | a |
| 3 | 21 | 97.2 | (± 1.22) | a |
| 3.5 | 20 | 97.5 | (± 1.74) | a |
| 4 | 20 | 96.2 | (± 2.21) | a |
| 4.5 | 21 | 97.5 | (± 1.66) | a |
| 5 | 13 | 95.9 | (± 8.77) | a |
| 6 | 8 | 102.9 | (± 4.16) | a |
| 7 | 6 | 135.5 | (± 33.84) | b |
| 8 | 5 | 132.6 | (± 34.00) | b |

[1] shared letters indicate groups that are not significantly different.

**Table 2. ANOVA statistics of Z score test.**

| Source | SS | df | MS | F | P value |
|---|---|---|---|---|---|
| Between Groups | 13791.628 | 8 | 1723.9536 | 10.093844 | $1.383 \times 10^{-10}$ |
| Within Groups | 19641.1457 | 115 | 170.792571 | | |
| Total | 33432.7742 | 123 | 271.811172 | | |

Samples that fell into the range of 98–102 consisted of 53.2% of the total samples. If the range was extended to 95–105, it included 77.0% of the samples. The *Ae. aegypti* distribution did not significantly deviate from normality (Shapiro-Wilk p = 0.699, Fig 4B) but also has a leftward skew. Overall, 47.6% of the samples fell in the range of 98–102 whereas 73.8% fell into the extended range of 95–105.

In this context, the contribution to precision of the larvae below the sensor that remain in the capillary can be considered. Based on our observations, they contribute approximately 1.4% to between-dispensing run deviation from the target when the target number to dispense is 100 and flushing is not performed between runs. Performing flushing requires emptying the reservoir and running clean water through the system or replacing the active feed components.

## Discussion

The principles of the APC are similar to the larva dispenser reported by Mamai *et al.* (2019) [12] called the MLC-CH1 v 1.0 developed by Radiation General Ltd. which uses optical density to trigger counts, is gravity fed and uses a pinch valve to control the flow. A fairly detailed description of the MLC-CH1 v 1.0 is provided in the manuscript. Using different testing methods on their device, the investigators reported an accuracy of ca. -7.0%. and high sensitivity to larval density, observing significant decreases in accuracy above 10 larvae per ml. Although we used larvae per second—different metric from Mamai *et al.* to indicate density assuming a constant water flow rate, it is possible to estimate a value of feed rates for comparison. They reported that the MLC-CH1 v 1.0 required 83 seconds to drain 400 ml or a flow rate of 4.8 ml/sec which is six times higher than the 0.77 ml/sec of the APC, and given the description of the device, it appears it uses a larger diameter detection tube than the APC. We cannot say whether this might contribute to the differences in the effect of larval density on accuracy. Another

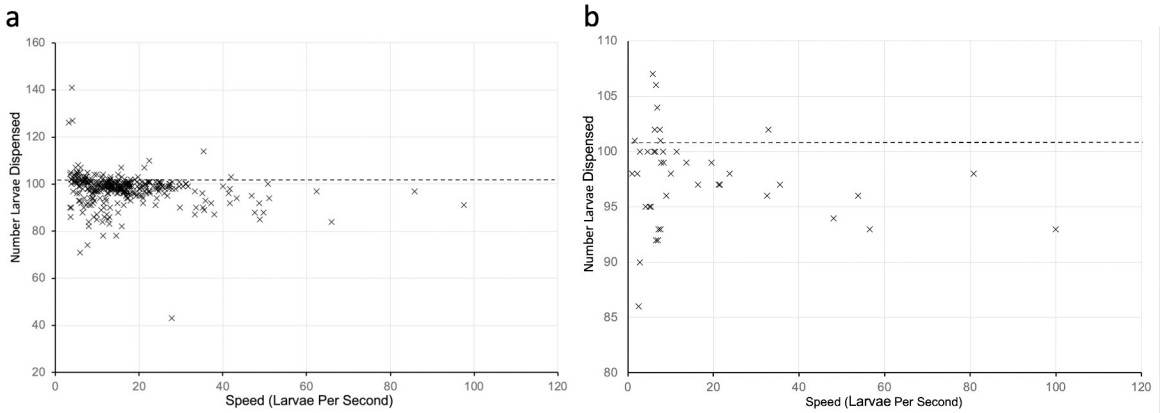

**Fig 3. Relationship between larval rate and number dispensed.** The relationship between the number of larvae per second passing the sensor and the number of (a) *An. gambiae* and (b) *Ae. aegypti* dispensed is shown.

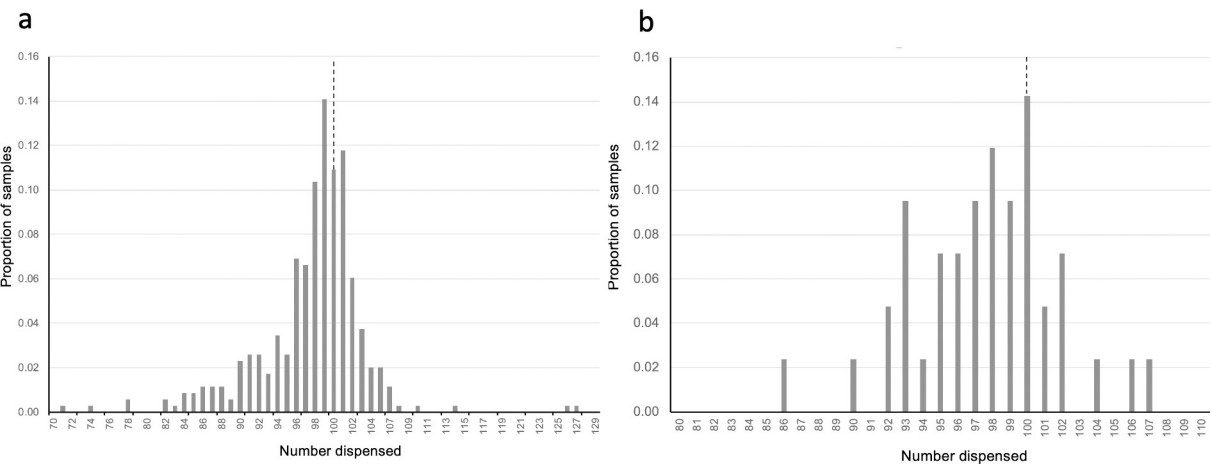

**Fig 4. Precision of dispensing.** The proportion of samples of various amounts dispensed when the target number was 100, indicated by the vertical dashed line. (a) The proportions of *An. gambiae* samples excluding two outlier samples with values of 40 and 141, consisting of a proportion of 0.003 of the samples. (b) The proportion of the number of samples of *Ae. aegypti* dispensed when the target number was 100.

dispensing device [16] is cited by Mamai et al (2019), but we have been unable to obtain this manuscript from Mamai or numerous library services using their reference information, nor have we located the journal by web searches. A third device developed by Orinno Technology has been noted in a news article [17] and presumably uses the same characteristic, but we believe a description of the device and its performance has not been published beyond the news article.

Somewhat different dispensing error rates by the APC might be expected for larger numbers of larvae. In a small preliminary trial counting 200 *An. gambiae* larvae (n = 19), we observed and average of 208 larvae (± 5.57) per dispensing. This contrasts with the under-dispensing observed with counts of 100. Therefore, users should determine the error rate that they experience with the specific numbers they dispense.

Potential users of the APC should keep in mind that there are thousands of combinations of settings that can be used to operate the APC. We make no claim that we have accomplished the highest level of performance that is possible, but we believe the ones we chose are acceptable for the purpose we had in mind; routine rearing in which predictable development rates and survival are important. It is possible to run small numbers of particles in an analytical mode and, by studying the tabular and graphical output, determine conditions that might improve performance e.g. how many measurements above the threshold trigger a count, how many decreasing values should pass before allowing a subsequent count.

The 1 mm ID capillary such as the APC is equipped with appears to be the smallest that is useful at the limit of gravity flow systems that do not have more head pressure. We performed preliminary trials with 0.75 mm ID capillaries in an attempt to improve accuracy by separating larvae more so than the standard 1 mm ID capillary, but the water came out in droplets rather than a continuous stream. We reckoned that this would result in a loss of accuracy and precision and did not investigate smaller capillaries further.

Because the APC requires setup and breakdown time whereas manual counting can be initiated with almost none of these, we estimated the amount of time required for either activity in a small preliminary trial. The time required to dispense 20 trays of 250 with the APC was 21 min 10 sec which includes 3 min 39 sec of setup and breakdown time. Manual counting of nine trays of 250 larvae by three people counting three trays each required an average of 3 min 18 sec per tray (range 2 min 12 sec– 4 min 10 sec) with essentially no setup and breakdown

time. Because setup and breakdown time is fixed, regardless of the number of trays dispensed, dispensing larger numbers of trays using the APC reduces the time per tray. However, if one is counting even two trays, it requires slightly less time to perform the dispensing using the APC (5 min 24 sec) over manual counting (6 min 36 sec). Simplistically, the break-even point to choose between the two methods is two trays, but the directness of manual counting will possibly cause most users to use the APC only for higher numbers of trays."

Every practical method of counting has inherent errors–including manual methods that are usually used as the reference, however, this method was the one we considered most reliable with which to compare the APC counts. When larvae are active in water samples rather than visualized in a photograph, the frequency of inaccuracies is likely to be even higher, but this is often not considered as a source of error.

Based on our experience, we make the following recommendations to improve the APC performance: (1) check the calibration frequently to ensure that the number of baseline standard deviations is < 18; (2) if used for mosquitoes or other samples that contain contaminants, take measures to reduce these by providing yeast or a dense, easily decanted food when hatching larvae, and by removing egg shells after hatching; (3) keep the capillary/sensor area dry to avoid getting it wet and causing damage (a spare sensor is provided with the device and can be changed easily by the user). Using a slightly longer capillary than is recommended helps assure a good seal to prevent leaking; (4) given that it requires some time to set up and flush the system, we believe it is best suited to situations in which approximately 3 or more trays of the same species or strain are needed rather than one or two trays of numerous different strains; and (5) maintain the water level at least half full and keep the larva feed rate below 50 LPS. Although we did not detect effects of higher feed rates, a degradation of performance seems likely due to overlapping larvae.

Here, we presented a series of trials to test the accuracy and precision of the APC for routine laboratory mosquito larva counting. In our hands, the APC is a useful device that requires little training and that is likely to improve not only mosquito culture, but may find use for other applications where regulated dispensing and counting are required. This device is expected to improve the accuracy and precision compared to the visual estimation methods that we have used in the past.

## Supporting information

**S1 Fig. Egg hatching arrangement.** In order to eliminate egg shells, *An. gambiae* embryos were placed on filter papers raised slightly above the water level on cloth sponge disc platforms. After hatching, larvae wriggle off the paper into the water and the filter paper can be lifted out, completely removing the shells. The larvae were then collected on a fine screen and the food particles decanted away with two or three rounds of slow pouring. It is important to use a sponge cloth that is smooth because textured cloths allow the larvae to collect on top of the sponge rather than wriggling into the water.
(DOCX)

**S2 Fig. Dispensed larvae.** Larvae were dispensed into small plastic cups and were transferred using a vacuum filter apparatus onto 15 cm diameter filter paper discs with the dispensing run sample identification number and target number written on it before being photographed.
(DOCX)

**S3 Fig. Details captured for the individual runs.** This capacity allows the user to recall the specific conditions that were used. The data is output in a comma delimited file (CSV) format and can be downloaded and stored in e.g. Microsoft Excel. These data were observed using the

web browser interface via a WiFi connection.
(DOCX)

**S4 Fig. Graph of data collected in diagnostic mode.** The peaks (lower section) are the amount of light sensed below the baseline. The method being used here is the 'Smoothed Z-Score'. Blue bars and points indicate all recorded absorbances above the Z score, whereas red points and bars indicate time points at which a 'count' is tabulated. Toward the right is a peak in which two larvae were counted that passed the detector in close succession. The sensor data can be collected in a detailed "diagnostic mode" or normal mode; the latter allows faster processing while the former can be used to develop parameters for the settings using the specific samples to be dispensed but requires more processing time and is not practical for dispensing runs of > 10 particles. Both modes create records of the settings for each dispensing run. While diagnostic mode data is not necessary for routine use, it can be helpful for testing the device as we did here and for establishing the proper conditions for routine operation. These can be visualized on a computer using a Wifi connection to the APC and can be transferred in CSV file format file for import into e.g. Microsoft Excel.
(DOCX)

## Acknowledgments

We appreciate the cooperation of Dr. Kyle Gabriel of Applied Scientific Solutions (AppliedScientificSolutionsLLC@gmail.com) who collaborated with the authors to develop the APC but who had no influence over the trial methods, the analysis of the data or the conclusions drawn by this study.

The findings and conclusions in this report are those of the authors and do not necessarily represent the official position of the Centers for Disease Control and Prevention. Use of trade names is for identification only and does not imply endorsement by the Centers for Disease Control and Prevention, the Public Health Service, or the U.S. Department of Health and Human Services.

The following reagents were obtained through the NIH Biodefense and Emerging Infections Research Resources Repository, NIAID, NIH: *An. gambiae*, strain 'G3' (MRA-112) and *Ae aegypti* 'New Orleans' strain (NR-49160).

## Author Contributions

**Conceptualization:** Mark Q. Benedict, Ellen M. Dotson.

**Data curation:** Mark Q. Benedict.

**Formal analysis:** Mark Q. Benedict.

**Funding acquisition:** Mark Q. Benedict, Ellen M. Dotson.

**Investigation:** Mark Q. Benedict, Priscila Bascuñán, Catherine M. Hunt, Erica I. Aviles, Rachel D. Rotenberry.

**Methodology:** Mark Q. Benedict, Catherine M. Hunt, Erica I. Aviles, Rachel D. Rotenberry.

**Project administration:** Mark Q. Benedict, Ellen M. Dotson.

**Resources:** Mark Q. Benedict, Priscila Bascuñán, Ellen M. Dotson.

**Supervision:** Mark Q. Benedict, Ellen M. Dotson.

**Validation:** Priscila Bascuñán, Erica I. Aviles, Rachel D. Rotenberry.

**Visualization:** Mark Q. Benedict, Priscila Bascuñán.

**Writing – original draft:** Mark Q. Benedict, Priscila Bascuñán, Catherine M. Hunt, Ellen M. Dotson.

**Writing – review & editing:** Mark Q. Benedict, Priscila Bascuñán, Catherine M. Hunt, Ellen M. Dotson.

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
