## [Decision Letter · Decision Letter 0]

1 Sep 2020

PONE-D-20-24168

Trials of the Automated Particle Counter for laboratory rearing of mosquito larvae

PLOS ONE

Dear Dr. Benedict, hope you are well and healthy!

Thank you for submitting your manuscript to PLOS ONE. After careful consideration, we feel that it has merit but does not fully meet PLOS ONE’s publication criteria as it currently stands. Therefore, we invite you to submit a revised version of the manuscript that addresses the points raised during the review process.

Reviewers still think that there are some minor needed modifications to your manuscript before being acceptable. Please try to respond to all queries raised here.

We look forward to receiving your revised manuscript.

Kind regards,

Luciano Andrade Moreira, PhD

Academic Editor

PLOS ONE

Journal Requirements:

[Funding Disclosure Statement: MQB, PB, CMH, RDR and EIA are paid employees of Target Malaria, a project that receives core funding from the Bill & Melinda Gates Foundation and from the Open Philanthropy Project Fund, an advised fund of Silicon Valley Community Foundation to the Target Malaria project.]

 [The funds for this study were awarded to the Foundation for the Centers for Disease Control and Prevention.]

4. Please upload a copy of Figure 6, to which you refer in your text on page 14. If the figure is no longer to be included as part of the submission please remove all reference to it within the text.

Reviewers' comments:

Reviewer's Responses to Questions

**Comments to the Author**

1. Is the manuscript technically sound, and do the data support the conclusions?

Reviewer #1: Yes

Reviewer #2: Yes

2. Has the statistical analysis been performed appropriately and rigorously? 

Reviewer #1: Yes

Reviewer #2: Yes

3. Have the authors made all data underlying the findings in their manuscript fully available?

Reviewer #1: Yes

Reviewer #2: Yes

4. Is the manuscript presented in an intelligible fashion and written in standard English?

Reviewer #1: Yes

Reviewer #2: Yes

5. Review Comments to the Author

Reviewer #1: Overview

In this paper draft, Benedict et al reviews the performance of a commercially available counter for L1 mosquito larvae. The authors give a broad overview of the device and parameters that can be adjusted when operating the device. The authors test the accuracy of the counter using two morphological different mosquito species and determined that the counter is accurate enough to use for routine rearing in a laboratory setting. The paper is well written, in clear English and easy to follow. The experimental design is sufficiently robust to support the results obtained and the discussion is supported by the observed results. Within this context, I think the manuscript meets the criteria for publication.

I do, however, have some concerns with the scope of testing and whether the authors have done enough to provide potential users of the system with the data required to make an informed choice on the whether the unit would provide a productivity improvement. I have some suggestions for additional testing that would give the manuscript more relevance to a wider audience.

Major Questions/Suggestions

1. Can the authors give some details on what the other algorithms are that are available and why one the Smoothed Z score was selected for testing?

2. What is the productivity break point for small scale rearing using the counter vs manual aliquoting? I think the authors should provide some additional data comparing the speed and accuracy of the counter against a human operator. At which point does it become more productive to use the counter vs just manual aliquoting? For instance, if I am rearing ten trays at a density of 250 larvae per week, does it make more sense to use the counter or should I stick with a technician?

3. Applicability for large scale rearing? Typically, L1 counters become much more attractive options when the scale of rearing increases. For large scale rearing facilities, manual aliquoting of L1 larvae is not feasible and I think it is important to demonstrate that the counter is still accurate when aliquoting larger (>1000) batches of larvae. In addition, the speed of operation is just as crucial, and the authors should include a comparison table to show the total time required to aliquot a range of L1 larvae.

Minor questions/issues

1. Line 233-235: Best Z-Score in Table 1 is 2.5 – why did the authors settle on 3.5? Can you provide a bit more context please?

2. Line 285: There is no Fig 6

Reviewer #2: Benedict et al report trials of a flow-cytometer type device called the ‘Automated Particle Counter’ (APC) to count mosquito larvae. Through tests of a number of key parameters, including Z score, accuracy and precision and larvae survivorship, they demonstrated APC may serve as an useful tool to control mosquito larvae density, an essential step in mosquito mass rearing. These results will facilitate the current effort to improve the effectiveness and quality control of mosquito mass rearing for developing novel mosquito controls strategy. Below are some minor points.

Figure 3 lists only the data of the measured value and LPS. How accuracy was calculated for the correlation test as indicated in the text (Line 250)? To be clear for the readers, the “accuracy” should be defined (degree of closeness of a measured value to its actual value?). It appears that data were enriched in low LPS and very few replicates was done at LPS over 60. Will this affect the conclusion?

Figure 4, “accuracy” and “precision” should be defined. Does this figure show only precision?

Line 268-290, what does the word “mode” mean? Why two different mode numbers were used for calculation of the accuracy?

It will be good to indicate the capacity of the APC in the discussion. For example, under an optimal condition, how much time is taken to calculate 1 million larvae or how many larvae can be dispensed in one hour?

Please indicate whether and how the Z score can be manually set for Automated Particle Counter in the method section.

6. PLOS authors have the option to publish the peer review history of their article (what does this mean?). If published, this will include your full peer review and any attached files.

Reviewer #1: No

Reviewer #2: No

---

## [Author Response · Author response to Decision Letter 0]

1 Oct 2020

See the 'response to reviewers' letter.

---

## [Decision Letter · Decision Letter 1]

16 Oct 2020

Trials of the Automated Particle Counter for laboratory rearing of mosquito larvae

PONE-D-20-24168R1

Dear Dr. Benedict,

We’re pleased to inform you that your manuscript has been judged scientifically suitable for publication and will be formally accepted for publication once it meets all outstanding technical requirements.

Kind regards,

Luciano Andrade Moreira, PhD

Academic Editor

PLOS ONE

Additional Editor Comments (optional):

Reviewers' comments:

Reviewer's Responses to Questions

**Comments to the Author**

1. If the authors have adequately addressed your comments raised in a previous round of review and you feel that this manuscript is now acceptable for publication, you may indicate that here to bypass the “Comments to the Author” section, enter your conflict of interest statement in the “Confidential to Editor” section, and submit your "Accept" recommendation.

Reviewer #1: All comments have been addressed

Reviewer #2: All comments have been addressed

2. Is the manuscript technically sound, and do the data support the conclusions?

Reviewer #1: Yes

Reviewer #2: (No Response)

3. Has the statistical analysis been performed appropriately and rigorously? 

Reviewer #1: Yes

Reviewer #2: (No Response)

4. Have the authors made all data underlying the findings in their manuscript fully available?

Reviewer #1: Yes

Reviewer #2: (No Response)

5. Is the manuscript presented in an intelligible fashion and written in standard English?

Reviewer #1: Yes

Reviewer #2: (No Response)

6. Review Comments to the Author

Reviewer #1: The authors addressed my comments adequately I believe the paper will be of interested to the wider mosquito research community. I recommend for the paper to be published.

Reviewer #2: (No Response)

7. PLOS authors have the option to publish the peer review history of their article (what does this mean?). If published, this will include your full peer review and any attached files.

Reviewer #1: No

Reviewer #2: No

---

## [Editor Report · Acceptance letter]

29 Oct 2020

PONE-D-20-24168R1 

Trials of the Automated Particle Counter for laboratory rearing of mosquito larvae 

Dear Dr. Benedict:

I'm pleased to inform you that your manuscript has been deemed suitable for publication in PLOS ONE. Congratulations! Your manuscript is now with our production department. 

Kind regards, 

on behalf of

Dr. Luciano Andrade Moreira 

Academic Editor

PLOS ONE